# Comment on "Comparison of Cloud Cover Detection Algorithms on Sentinel-2 Images of the Amazon Tropical Forest"

**Olivier Hagolle * and Jerome Colin**

Centre d'Études Spatiales de la Biosphère, CESBIO Unité mixte Université de Toulouse-CNES-CNRS-INRAE-IRD, 18 Avenue E.Belin, 31401 Toulouse CEDEX 9, France; jerome.colin@cesbio.cnes.fr
* Correspondence: olivier.hagolle@cesbio.cnes.fr; Tel.: +33-561282135

**Abstract:** In their recent study, Sanchez et al. compared various cloud detection methods applied to Sentinel-2, specifically on images acquired over the Amazonian region, known for its frequent cloud cover. Comparison of cloud screening methods for optical satellite images is a complex task, which must take several parameters into account, such as the definition of a cloud, which can differ according to the methods, the different coding of the cloud and shadow masks, the possible dilation of masks, and also the way the method must be used to perform in nominal conditions. We found that the otherwise serious and useful comparison of cloud masks by Sanchez et al. is not fair to the real performances of MAJA cloud detection, for two reasons: (i) two thirds of the images used in the comparison were acquired before the launch of Sentinel-2B satellite, when the revisit of the Sentinel-2 mission was 20 days instead of five days for the nominal conditions of the mission, and (ii) there is an error in the understanding of how MAJA cloud masks are coded which also probably artificially degraded the results of MAJA as compared to the other methods.

**Keywords:** Sentinel-2; cloud mask; cloud shadow; validation; MAJA

In their study over Amazonia, Sanchez et al. [1] found that the performances of MAJA cloud screening software are not as good as those obtained in a similar study by Baetens et al [2]. A large part of the discussion in the article was dedicated to explaining the differences. It concluded that they are probably related to the specificities of cloud cover and surface reflectances in Amazonia. The authors also noticed that MAJA (MACCS-ATCOR Joint Algorithm) was surprisingly not able to detect cloud shadows in Amazonia.

While most cloud detection methods process successive images independently, our method for cloud detection and atmospheric correction, MAJA, uses the generally slow variations of surface reflectances as a function of time to better detect the clouds and their shadows [3]. Our cloud detection method uses time series of images as input, processes the images in chronological order, and the quality of its results improves when the interval between successive observations of the land surface is reduced. In their study, Sanchez et al. used twenty manually classified images taken over five tiles scattered in the Amazonia (Table 2 of their paper). Thirteen of these images were acquired by Sentinel-2A, before Sentinel-2B was put in operation in July 2017. Moreover, until November 2017, the revisit of each Sentinel-2 satellites was 10 days in Europe and Africa, and 20 days on the other continents. As an example, Table 1 shows the number of images acquired by Sentinel-2A or Sentinel-2B over an Amazonian tile used in the Sanchez et al. study, for the same three-month period from 2016 to 2020.

MAJA has been designed to perform optimally in the nominal conditions of Sentinel-2, with a revisit of five days from the same path. For most images in the time series used by Sanchez et al., the revisit was 20 days. Given the frequent cloud cover in Amazonia, MAJA does not always have sufficient cloud free observations of the surface, and works with a fallback mono-temporal method to detect the clouds and shadows. Sanchez et al.

do not provide validation statistics per image, but Table 5 of Sanchez et al. provides results per tiles, accumulating the four available dates for each tile. The two tiles which have only dates in 2016 and 2017 (T20NPH and T22NCG) exhibit the worst performances for MAJA, which shows that MAJA's performance should be much better for the nominal configuration of Sentinel-2 with a revisit of five days.

**Table 1.** Number of images gathered between first of April and first of July for the MGRS tile T22MCA used in Sanchez et al.

| Year | 2016 | 2017 | 2018 | 2019 | 2020 |
|---|---|---|---|---|---|
| Number of images | 4 | 5 | 18 | 18 | 18 |

Moreover, we found that the authors wrongly interpreted MAJA's cloud mask. For each pixel, MAJA not only provides a binary valid/invalid mask, but the mask is expressed as bit mask which tells which of MAJA's tests declared a pixel invalid. Such a representation is a little complex but has the advantage to provide a complete information on a small volume (one byte per pixel). Table 3 of the original paper provides a conversion table for MAJA cloud mask which is partly false, for instance, only the value 0 in MAJA cloud masks correspond to a cloud free/shadow free pixel while Table 3 lists 0 and 1. Two figures of the Sanchez et al. paper (the graphical abstract and Figure 3) show that the shadows detected by FMASK are also classified as invalid in MAJA, but the authors wrongly indicated that MAJA classifies them as clouds instead of shadows. For the sake of comparison, we provide in Figure 1 the MAJA cloud mask for the same tile and date as the Figure 3 in Sanchez et al. with the correct representation of the cloud bit-mask. We understood that, in Sanchez et al., the classification performances have been computed for three classes (clear, cloud, shadow), and, in that case, the wrong interpretation of MAJA mask values must also have a large impact on the performances. In addition, we are not sure it is optimal to separate cloud and cloud shadows in distinct classes for validation purposes, as shadows on other clouds and mask dilation process lead to mixed classes. End-users usually expect a mask to discriminate between valid and invalid pixels with the highest possible accuracy, whether invalid ones are clouds or cloud shadows should remain a second order performance criterion.

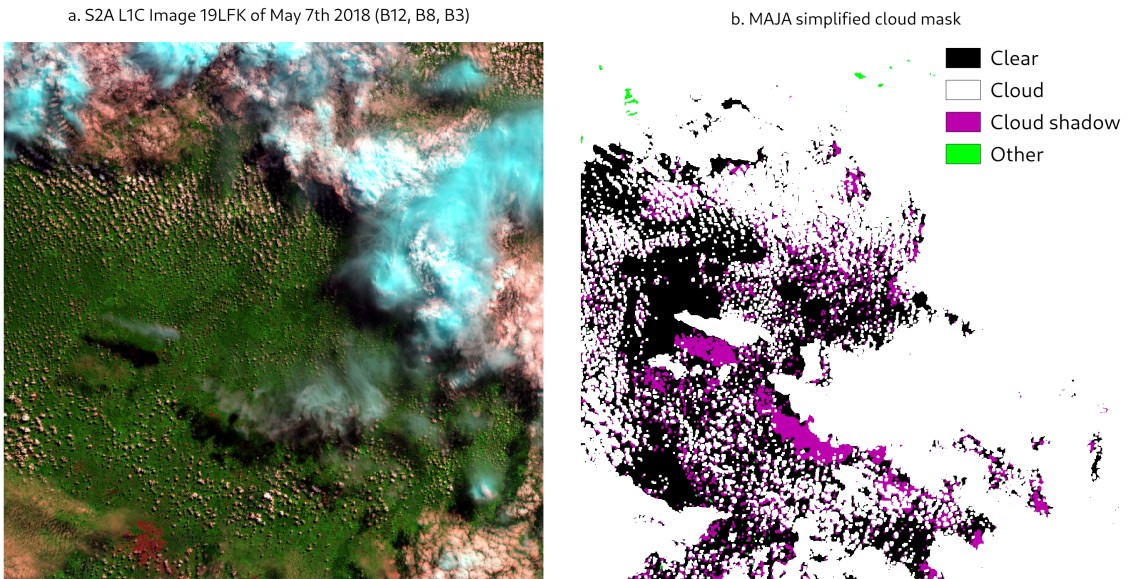

**Figure 1.** Equivalent to Sanchez et al. Figure 3 with: (**a**) the SENTINEL-2A L1C image of 7 May 2018 and (**b**) the MAJA cloud mask with a proper representation of binary codes. Here, we aggregated bit-mask combinations to match the classes as shown in Sanchez et al.

The study of Sanchez et al. is a very interesting initiative, since the Amazonian regions pose specific challenges to cloud masking, as the cloud cover is usually high and the surface reflectances are low. In such regions, the multi-temporal method, which is a corner stone of MAJA, partly loses its advantages as the surface is seldom visible and reflectances can significantly evolve from one cloud free observation to the next. Even if Amazonia is not a use case where MAJA will obtain its best results, we think that the two shortcomings detailed above have resulted in performances for MAJA that are not the nominal ones for Sentinel-2, even over Amazonia.

**Author Contributions:** Writing O.H., J.C.; analysis J.C., O.H. Both authors have read and agreed to the published version of the manuscript.

**Funding:** This research received no external funding.

**Institutional Review Board Statement:** Not applicable.

**Informed Consent Statement:** Not applicable.

**Data Availability Statement:** The study did not report any data.

**Conflicts of Interest:** The authors declare no conflict of interest.

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
