# Peer review of "Comment on “Comparison of Cloud Cover Detection Algorithms on Sentinel-2 Images of the Amazon Tropical Forest”"

_remotesensing, doi:10.3390/rs13051023_

Round 1

Reviewer 1 Report

I agree with the comment of Hagolle and Colin.

It is very difficult in an objective way to compare different cloud detection methods  when  given input parameter values and cases affect more or less to the results. An experimented user may get better results by knowing how to tailor the parameter values for different input cases, like  satellite image type or geographical area or land cover.

It is justified to give the comment.

------------------

Line 40. Revisit

Author Response

We wish to thank the reviewer for the careful reading, his/hers appreciation of our comment and the suggestion regarding the typo line 40.

Reviewer 2 Report

The comments made by Hagolle and Colin are clear and relevant. They perfectly underline the difficulties in appropriating and mastering different tools, especially if they do not have the same objectives. Moreover, it seems useful to me to add the remarks on the availability of Sentinel-2 data at the beginning of the activity and in particular the dispersion of temporal data (Table 1). All these remarks should be associated with the original paper by Sanchez et al to qualify the conclusion about the MAJA chain. Ideally, Sanchez et al. would use the classification using MAJA with respect to the remark of the authors: "The Table 3 of the original paper provides a conversion table for MAJA cloud mask which is partly false, for instance, only the value 0 in MAJA cloud masks corresponds to a cloud free/shadow free pixel while table 3 lists 0 and 1. Two figures of Sanchez et al. paper (the graphical abstract and Figure 3) show that the shadows detected by FMASK are also classified as invalid in MAJA, but the authors wrongly indicated that MAJA classifies them as clouds instead of shadows." However, as the authors lucidly point out, performances with the MAJA channel would not be as high as with FMASK.

Author Response

We wish to thank the reviewer for the careful reading of our comment paper, and his/hers assesment.